# Prospective Multi-Institutional Observational Study of Retreatment with Anti-PD-1/PD-L1 Antibodies in Patients with Non-Small Cell Lung Cancer Previously Treated with Anti-PD-1/PD-L1 Plus Chemotherapy: NJLCG (North Japan Lung Cancer Group) Trial 1901

**DOI:** 10.3390/cancers17091551

**Published:** 2025-05-02

**Authors:** Shin Saito, Yosuke Kawashima, Hisashi Tanaka, Naruo Yoshimura, Yoko Tsukita, Ryota Saito, Taku Nakagawa, Minehiko Inomata, Hiromi Nagashima, Shunichi Sugawara

**Affiliations:** 1Department of Pulmonary Medicine, Sendai Kousei Hospital, Sendai 981-0914, Japan; saitou.meidai@gmail.com (S.S.); swara357@sendai-kousei-hospital.jp (S.S.); 2Department of Internal Medicine, Matsuzono Daini Hospital, Morioka 020-0103, Japan; 3Department of Respiratory Medicine, Hirosaki University Hospital, Hirosaki 036-8563, Japan; xyghx335@gmail.com; 4Department of Respiratory Medicine, Tohoku Medical and Pharmaceutical University Hospital, Sendai 983-8512, Japan; y-naruo@sc4.so-net.ne.jp; 5Department of Respiratory Medicine, Tohoku University Graduate School of Medicine, Sendai 980-8574, Japan; yoko.tsukita.b4@tohoku.ac.jp; 6Department of Respiratory Medicine, Yamanashi Prefectural Central Hospital, Kofu 400-8506, Japan; beambitious0529@yahoo.co.jp; 7Department of Thoracic Surgery, Omagari Kosei Medical Center, Daisen 014-0027, Japan; drtakubo@yahoo.co.jp; 8First Department of Internal Medicine, Toyama University Hospital, Toyama 930-0194, Japan; minomata@med.u-toyama.ac.jp; 9Department of Respiratory Medicine, Iwate Medical University Hospital, Iwate 028-3695, Japan; all-checker1983@m7.dion.ne.jp

**Keywords:** nivolumab, pembrolizumab, atezolizumab, immune-checkpoint inhibitor, non-small cell lung cancer, retreatment

## Abstract

The efficacy of retreatment with immune checkpoint inhibitor (ICI) monotherapy after progression to platinum-based chemotherapy plus ICI in non-small-cell lung cancer remains uncertain and needs to be prospectively assessed. We aimed to prospectively evaluate retreatment with ICI monotherapy (nivolumab, pembrolizumab, and atezolizumab) in patients with advanced non-small cell lung cancer who relapsed after platinum-based chemotherapy plus ICIs. This study reports an overall response rate of 10.5%, with some patients achieving long-term progression-free survival. Prognostic factors for a longer progression-free survival included a longer ICI-free interval (>11.9 months) and prior anti-programmed death ligand 1 therapy. The patient safety profile was also manageable. Further studies are needed to identify the patient subgroups that benefit from ICI retreatment.

## 1. Introduction

The emergence of immune checkpoint inhibitors (ICIs) has revolutionized standard therapies for non-small cell lung cancer (NSCLC). ICIs, including anti-programmed death 1 (anti-PD-1) and anti-programmed death ligand 1 (anti-PD-L1) inhibitors with platinum-based chemotherapy (chemo-ICI), have become the standard first-line treatments for metastatic and locally advanced NSCLC, as demonstrated in several randomized phase III trials [1,2,3]. However, the development of resistance to chemo-ICIs remains a major clinical challenge. Although ICI monotherapies (nivolumab, pembrolizumab, and atezolizumab) have shown clinical benefits after second-line treatment in patients with NSCLC without prior exposure to ICIs [4,5,6,7], standard second-line therapies following the progression of initial chemo-ICIs are limited to cytotoxic agents, such as docetaxel [8,9], which offer limited efficacy. Therefore, additional therapeutic options need to be developed.

Previous studies on ICI retreatment have mainly been retrospective, reporting limited yet heterogeneous outcomes, with overall response rates (ORRs) ranging from 0% to 22.5% [10,11,12,13,14]. This heterogeneity can be attributed to the diverse patient backgrounds, varying reasons for initial ICI failure, and different sequences of ICI therapies. A meta-analysis of retrospective studies suggested that progression following the discontinuation of the first ICI due to immune-related adverse events (irAEs) or ICI cessation is a prognostic factor for prolonged progression-free survival (PFS) after ICI retreatment [15]. However, possible retrospective bias highlights the need for prospective studies to confirm the efficacy and prognostic factors associated with ICI retreatment. Prior to this study, a prospective phase II trial (WJOG9616L) conducted by the West Japan Oncology Group investigated the efficacy of nivolumab monotherapy following the failure of prior ICI treatment [16]. In this trial, the ICI-free interval, defined as the period from the last administration of the first ICI to ICI retreatment, was the sole prognostic factor for ICI retreatment. However, given the scarcity of prospective studies evaluating ICI treatments, further research is necessary.

Additionally, while different types of ICIs (anti-PD-1 and anti-PD-L1) are available for both first- and later-line treatments, determining ICI regimens for retreatment based on prior ICI therapy remains a clinical challenge. Given the differences in the abilities of anti-PD-1 and anti-PD-L1 to inhibit signaling through programmed death ligand 2 (PD-L2), the use of anti-PD-1 after anti-PD-L1 therapy could theoretically be more beneficial. However, data validating ICI switching are limited. Therefore, we conducted a prospective observational study to assess the efficacy, safety, and prognostic factors of ICI retreatment after the failure of chemo-ICIs while focusing on the significance of the ICI sequence of anti-PD-1 and anti-PD-L1 therapies (North Japan Lung Cancer Group Trial: 1901 [NJLCG 1901]).

## 2. Materials and Methods

### 2.1. Study Design and Patient Eligibility

The NJLCG 1901 study was a multi-institutional, prospective observational study conducted at the institutions of the NJLCG. Eligible patients had histologically confirmed NSCLC with advanced disease or recurrence after curative treatment (surgery or chemoradiotherapy [CRT]), were negative for driver gene mutations, and had experienced disease progression (PD) after receiving platinum-based chemotherapy combined with either anti-PD-1 or anti-PD-L1. Patients who were treated with CRT followed by durvalumab and relapsed were included after protocol revision. Patients were required to have measurable disease according to the Response Evaluation Criteria in Solid Tumors version 1.1 and be at least 20 years of age. The exclusion criteria included active multiple malignancies, history of severe irAEs, active autoimmune diseases, and current use of steroids exceeding 10 mg/day of the prednisone equivalent.

### 2.2. Treatment, Assessment, and Endpoints

The patients were prospectively enrolled and treated with ICI monotherapy (nivolumab, pembrolizumab, or atezolizumab, as determined by the physicians at each institution) within 2 weeks. The treatment was continued until PD or intolerable adverse events (AEs) occurred. Computed tomography-based assessments were recommended every 6 to 12 weeks. AEs occurring within 1 month of the final dose of ICI retreatment were recorded. The primary endpoint was the ORR, as assessed by physicians at each institution. Secondary endpoints included PFS, overall survival (OS), disease control rate (DCR), and safety. The prognostic clinical factors for PFS were also investigated. The data cutoff date was 29 February 2024.

### 2.3. Statistical Analysis

The ORR and DCR were calculated with 95% confidence intervals (CIs). To examine the relationship between the ICI-free interval and response, the ORR and DCR were also calculated by stratifying patients into two groups based on whether their ICI-free interval was longer or shorter than the median. PFS and OS were estimated using the Kaplan–Meier method. The association between PFS after ICI retreatment and clinical factors was assessed using univariate analysis with a Cox proportional hazards model. The clinical factors included histological type, metastatic site, tumor proportion score (TPS), and type of prior ICI treatment, as well as exploratory factors such as the ICI-free interval, reasons for discontinuation of chemo-ICIs (due to PD or other reasons [non-PD: toxicity or completion of durvalumab]), serum albumin level, and neutrophil-to-lymphocyte ratio. Continuous variables were divided into two groups using the median as a cutoff. Factors with a *p*-value of less than 0.05 in the univariate analysis were further examined in the multivariate analysis. Statistical significance was set at a *p*-value of less than 0.05. All statistical analyses were performed using EZR (Saitama Medical Center, Jichi Medical University, Saitama, Japan), a graphical user interface for R (R Foundation for Statistical Computing, Vienna, Austria) [17].

### 2.4. Sample Size

The standard therapy administered to the study population was cytotoxic chemotherapy, with a threshold response rate of 8%. A response rate of 20% was expected with ICI retreatment. Based on a significance level of an α of 0.05 (two-sided) and power of 80%, 63 patients were required.

### 2.5. Ethical Considerations

This study was conducted in accordance with the ethical standards of the Declaration of Helsinki, and the study protocol was approved by the institutional review boards of all participating institutions. All participants provided written informed consent prior to enrollment. The study was registered with the University Hospital Medical Information Network of Japan (registration number: UMIN000038413).

## 3. Results

### 3.1. Patient Characteristics

Due to delays in patient recruitment, the intended sample size was not reached. Ultimately, 40 patients were enrolled across eight institutions in this trial between August 2020 and November 2023. Two patients were excluded from the analysis: one did not receive ICI retreatment after registration, and the other was found not to have a target lesion after registration. Baseline patient characteristics are summarized in Table 1. Among the 38 patients analyzed, all had a performance status (PS) of 0–1, and 24 (63.2%) had non-squamous histology. Four patients (10.5%) who relapsed after CRT and durvalumab therapy were included in this study. The numbers of patients receiving ICI retreatment with nivolumab, pembrolizumab, and atezolizumab were 21 (55.2%), five (13.2%), and 12 (31.6%), respectively. The median number of treatment lines administered before ICI retreatment (median [range]) was 2.5 (1–4). The median ICI-free interval (median [range], months) was 11.9 (1.0–38.9). Thirty-three patients (86.8%) had a history of docetaxel treatment. Regarding the reasons for discontinuation of prior chemo-ICIs, 29 patients (76.3%) discontinued treatment owing to progression, whereas nine patients (23.7%) discontinued treatment for reasons other than progression: toxicity (*n* = 8) and completion of durvalumab (*n* = 1).

### 3.2. Efficacy

The results of the primary endpoint are shown in Table 2. The best tumor reduction is displayed as a waterfall plot, along with individual patient characteristics in Figure 1. Among the 38 patients, 1 achieved a complete response (CR), and 3 achieved a partial response (PR). Fourteen patients had stable disease (SD). Thus, the ORR and DCR were 10.5% (95% CI: 2.9–24.8%) and 47.4% (95% CI: 31.0–64.2%), respectively. Three patients were not assessed for best response because of death prior to the first tumor assessment. Among the four responders, three were treated with nivolumab and had a history of docetaxel treatment. Two responders discontinued prior chemo-ICIs owing to PD, whereas the other two discontinued owing to reasons other than PD (one with CR had completed a 1-year durvalumab treatment, and the other discontinued chemo-ICIs owing to toxicity). The median time to best response was 2.4 (range: 2.1–4.7) months. The ICI-free intervals of four responders were longer than the median. Among those with the ICI-free interval longer than 11.9 months, the ORR and DCR were 21.1% (95% CI: 6.1–45.6%) and 63.2% (95% CI: 38.4–83.7%). There were no responders among patients with an ICI-free interval ≤ 11.9 months.

The Kaplan–Meier curves for PFS and OS after ICI retreatment are shown in Figure 2. After a median follow-up of 8.1 months, 32 (84.2%) and 24 (63.2%) patients experienced PFS and OS events, respectively. The median PFS was 2.5 months (95% CI: 1.6–4.4), and the 1- and 2-year PFS rates were both 13.8% (95% CI: 5.1–26.8%). The median OS was 9.9 months (95% CI: 8.0–13.6). The PFS duration and patient characteristics are summarized in the Swimmer plot in Figure 3. Six patients achieved a PFS longer than 6 months; all these patients had a history of docetaxel treatment, and five did not experience PD at the data cut-off.

Table 3 shows the results of the univariate and multivariate analyses of the prognostic factors for PFS after ICI retreatment. In the univariate analysis, non-squamous histology, an ICI-free interval (longer than 11.9 months), and a prior history of anti-PD-L1 treatment were associated with longer PFS. Multivariate analysis of these three factors reveals that the ICI-free interval (>11.9 months) and prior anti-PD-L1 treatment were significant prognostic factors for PFS. To evaluate the prognostic impact of prior anti-PD-L1 therapy, the relationship between the sequence of prior and retreatment ICI therapies and PFS was explored as an exploratory analysis (Figure 4). The ICI sequences consisted of 12 cases of prior anti-PD-1 treatment followed by anti-PD-L1 retreatment, 10 cases of prior anti-PD-L1 treatment followed by anti-PD-1 retreatment, and 16 cases of anti-PD-1 for both prior and retreatment therapies. Regarding ICI switching, the sequence of anti-PD-1 retreatment after anti-PD-L1 showed a tendency toward longer PFS than reverse switching. The Kaplan–Meier curves for PFS according to the other key subgroups are shown in Figure 5. The reasons for prior chemo-ICI failure (PD vs. non-PD, toxicity, or durvalumab completion) and TPS were not associated with statistically significant differences in PFS. To explore the potential impact of including patients with recurrence after chemoradiotherapy and durvalumab, univariate and multivariate analyses of prognostic factors for PFS after ICI retreatment, excluding these patients, were conducted (Appendix A). This exploratory analysis demonstrates trends consistent with those observed in the main analysis. As an additional note, all nineteen patients with an ICI-free interval longer than 11.9 months received ICI retreatment after third-line therapy: five patients had two prior cytotoxic treatment lines, nine had three, and five had four or more. No statistically significant difference in PFS after ICI retreatment was observed based on the number of prior treatment lines in this subgroup (Appendix A).

### 3.3. Safety

Among the AEs that occurred from the start of ICI retreatment to 1 month after the last dose (Table 4), those reported as irAEs included rashes (*n* = 3; 7.9%, all grades) and colitis, pneumonitis, and myositis (*n* = 1; 2.6%). Grade 3 or higher irAEs were observed in rash and myositis, each occurring in 2.6% of patients. No serious AEs deemed unrelated to ICI therapy were reported. Only one case (2.6%) discontinued treatment because of an AE.

## 4. Discussion

In this prospective observational study, retreatment with ICI monotherapy after chemo-ICI resulted in an ORR of 10.5% and a DCR of 47.4%. Although the limited sample size in this study may have led to reduced statistical power to detect the ORR with ICI retreatment, the clinical benefit of ICI retreatment was limited in most patients, consistent with previous retrospective studies [10,11] and the prospective WJOG9616L trial. However, the standard treatment for the patients in this study would typically have been cytotoxic chemotherapy, such as docetaxel, with a relatively low ORR of approximately 20% in previous trials [18,19]. As 86.8% of patients in this trial received docetaxel before ICI retreatment, the observed ORR in heavily pretreated patients in the late-line setting is noteworthy. Although the median PFS was short (2.5 months), the 1- and 2-year survival rates were identical, with a tail plateau effect observed. The fact that a small subgroup of study patients achieved long-term PFS, even in the late-line treatment setting, confirmed the presence of a clinical benefit from ICI retreatment. This highlights the need for further investigation to identify patients who may benefit from ICI retreatment.

The prognostic factors related to longer PFS after ICI retreatment were a longer ICI-free interval (longer than 11.9 months) and prior anti-PD-L1 treatment. A longer ICI-free interval was also reported as the sole clinical factor for PFS in the WJOG 9616 L trial, which, to the best of our knowledge, is the only prospective study to investigate retreatment with ICI monotherapy in Japan, except for our trial. The ICI-free interval may be associated with prolonged PFS through several potential mechanisms. First, a break in ICI therapy could alleviate T-cell exhaustion, which is recognized as a mechanism of ICI resistance [20,21], thereby restoring immune cell function. Second, the effect of subsequent chemotherapy during the ICI-free period may have played a role in enhancing the efficacy of ICI retreatment. Several studies have reported that chemotherapy can increase PD-L1 expression, possibly by affecting the tumor microenvironment [22,23]. However, in this study, pathological evaluations and biomarker explorations were not conducted to assess changes in the tumor microenvironment due to cytotoxic chemotherapy administration during the ICI-free period. Finally, patients with a longer ICI-free interval may include a larger number of clinically stable patients, which may be associated with longer PFS after ICI retreatment. Patients with rapid tumor progression often require frequent regimen changes, which may result in shorter ICI-free intervals. Nonetheless, the fact that a longer ICI-free interval was associated not only with prolonged PFS but also with a numerically higher ORR suggests that the ICI-free interval is the most significant clinical factor for identifying patients likely to benefit from ICI retreatment. However, as noted above, the ICI-free interval encompasses a variety of clinical courses. It remains unclear whether the duration of the ICI-free interval itself contributes to overcoming ICI resistance or whether changes in the tumor microenvironment induced by intervening treatments are more critical. Further comprehensive investigations with tumor microenvironmental assessments are essential to elucidate the role of the ICI-free interval in modulating ICI resistance.

Prolonged PFS in patients with prior anti-PD-L1 treatment may be associated with a specific ICI sequence pattern. In an exploratory analysis of the ICI sequence, patients with anti-PD-1 retreatment among those with an anti-PD-L1 history exhibited longer PFS, with a higher proportion of patients showing a tail plateau. In contrast, retreatment with anti-PD-L1 after anti-PD-1 therapy did not result in clinical benefits. This finding suggests the effectiveness of switching from anti-PD-L1 to anti-PD-1 therapy. A distinguishing feature of PD-1 antibodies is their inhibitory effect on PD-L2, which is different from that of PD-L1 antibodies. Several studies have indicated a potential role of PD-L2 in ICI resistance [24]. Therefore, reversing the resistance by administering anti-PD-1 to patients who have only been treated with anti-PD-L1 alone is plausible. However, the PD-L1 treatment group in this trial also included patients who received durvalumab after CRT, indicating that other clinical factors affected the outcomes. Owing to the limited sample size of this study, performing an evaluation excluding all confounding factors would be difficult. However, the observation that similar trends were seen in the multivariate analysis of prognostic factors for PFS, even after excluding patients with recurrence following CRT and durvalumab, suggests that the sequence of ICI treatment is an important factor to consider. Currently, there is limited data on switching ICIs. Retrospective analyses reported the benefit of ICI switching in NSCLC; however, the sample size was small, and the order of anti-PD-1 and anti-PD-L1 was not defined [25,26]. Our study provides important prospective data comparing ICI switching, focusing on the benefits of switching from anti-PD-L1 to anti-PD-1. Additionally, although the treatment regimens were different, Mouri et al. reported the efficacy of nivolumab and ipilimumab as first-line treatments after chemoradiation plus durvalumab resistance in NSCLC, which also indicates the clinical benefit of ICI switching [27]. Although our study focused only on anti-PD-1 and anti-PD-L1 therapies, anti-cytotoxic T lymphocyte antigen 4 (CTLA-4) inhibitors have become mainstream options for first-line treatment [28,29,30]. Therefore, an evaluation of the optimal sequence that includes CTLA-4 inhibitors may be pertinent in the future. As for other prognostic factors, the discontinuation of ICIs other than PD, as reported in prior studies [15,31], was not reproduced in this trial. This could be owing to the limited number of cases in the non-PD discontinuation group and lack of statistical power. Furthermore, this trial did not limit the discontinuation of chemo-ICIs to irAEs but also included discontinuation owing to chemotherapy toxicity, which may have affected the negative results.

No serious AEs were reported regarding the safety profile of ICI retreatment. However, this study only collected information on AEs up to 1 month after the final ICI retreatment. Therefore, safety related to late-onset complications after treatment completion could not be assessed, which may lead to an underestimation of late-onset irAEs. Overall, ICI retreatment appeared to be an acceptable option from a safety perspective.

Our study has several limitations. First, the required sample size was not achieved, leading to limited statistical power and difficulty in conducting multivariate analyses with other possible clinical factors. Second, the follow-up period for PFS and OS was insufficient because the ORR was the primary endpoint. Third, the study did not include information regarding the recurrence patterns at the time of initial ICI resistance, particularly regarding whether the progression was due to oligometastasis. Finally, we did not perform biomarker analyses using tissue or blood samples.

## 5. Conclusions

After initial chemo-ICI, retreatment with ICI monotherapy showed limited but certain efficacy with long-term PFS in a subset of patients in the late-line setting. A longer ICI-free interval (than 11.9 months) and prior anti-PD-L1 treatment were identified as prognostic factors. Further large-scale investigations are needed to identify specific subgroups that may benefit from ICI treatment.

## Figures and Tables

**Figure 1 cancers-17-01551-f001:**
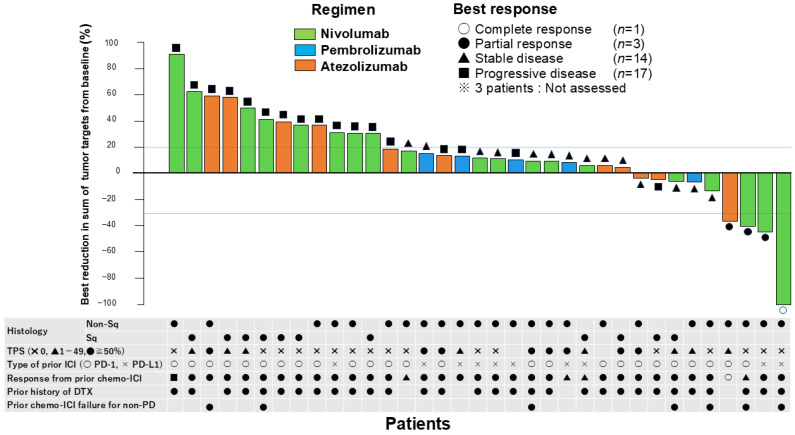
Waterfall plot of best tumor reduction and patient factors. Abbreviations: CR, complete response; PR, partial response; SD, stable disease; PD, progressive disease; Sq, squamous; TPS, tumor proportion score; ICI, immune checkpoint inhibitor; PD-1, anti-PD-1 (anti-programmed death 1); PD-L1, anti-PD-L1 (anti-programmed death ligand 1); DTX, docetaxel.

**Figure 2 cancers-17-01551-f002:**
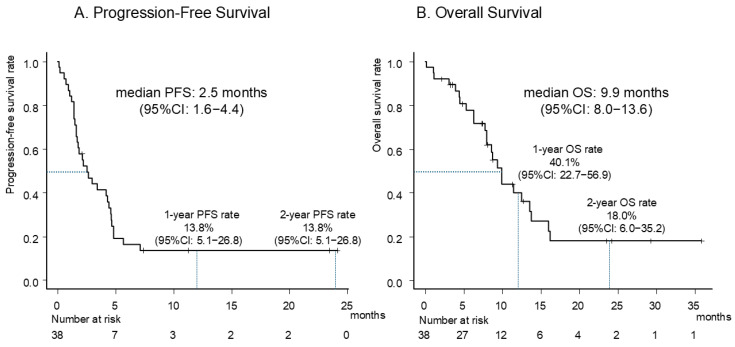
Kaplan–Meier curves of PFS (**A**) and OS (**B**) for ICI retreatment. Abbreviations: PFS, progression-free survival; OS, overall survival; ICI, immune checkpoint inhibitors; CI, confidence interval.

**Figure 3 cancers-17-01551-f003:**
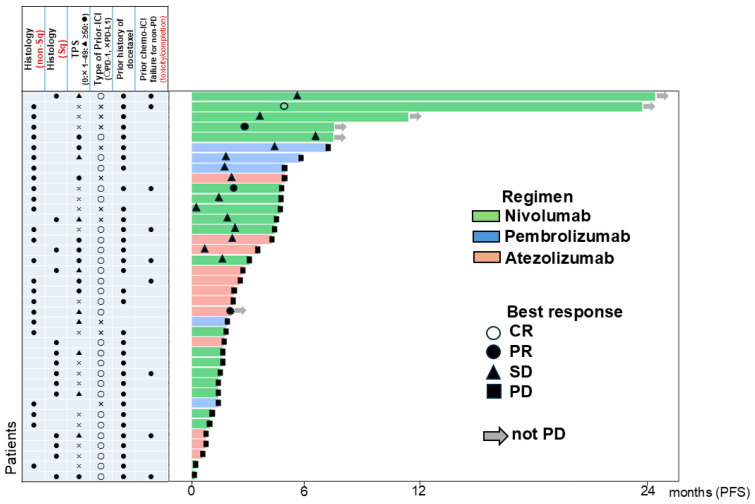
Duration of PFS on ICI retreatment and patient characteristics in Swimmer’s plot. Abbreviations: PFS, progression-free survival; ICI, immune checkpoint inhibitors; Sq, squamous cell carcinoma; TPS, tumor proportion score; CR, complete response; PR, partial response; SD, stable disease; PD, progressive disease.

**Figure 4 cancers-17-01551-f004:**
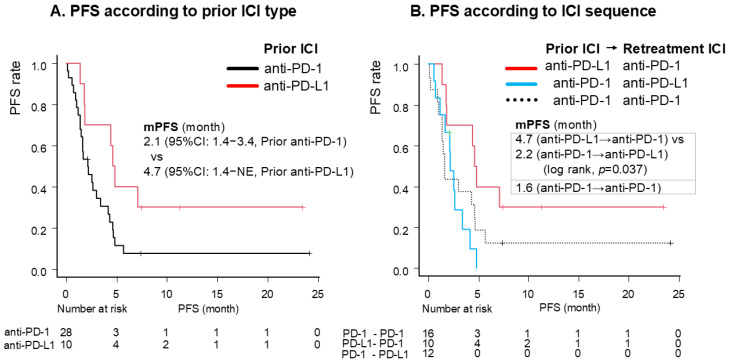
PFS according to the type of ICI sequences. PFS according to the type of prior ICI (**A**) and the type of prior and rechallenge ICI (**B**). The log-rank test was used to compare the switching between the anti-PD-L1 → anti-PD-1 group and the anti-PD-1 → anti-PD-L1 group. Abbreviations: PFS, progression-free survival; ICI, immune checkpoint inhibitor; mPFS, median progression-free survival; CI, confidence interval; NE, not estimable; anti-PD-1, anti-programmed death 1; anti-PD-L1, anti-programmed death ligand 1.

**Figure 5 cancers-17-01551-f005:**
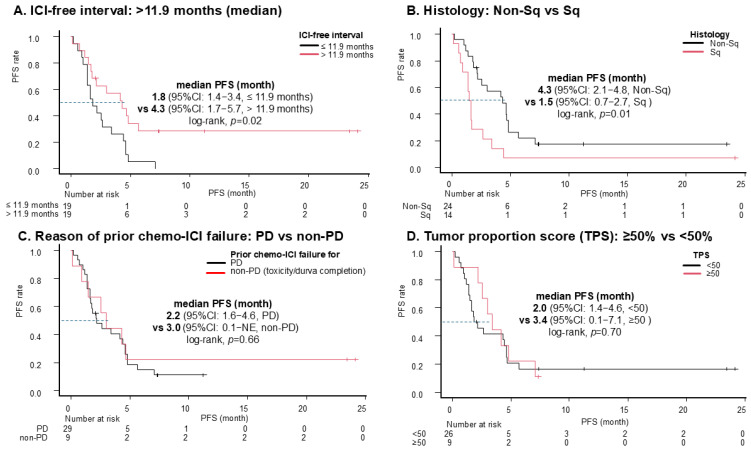
PFS for ICI retreatment according to key subgroups. ICI-free interval longer than the median (11.9 months) or not (**A**), histology (**B**), prior chemo-ICI failure for PD or not (**C**), TPS ≥ 50% or not (**D**). Abbreviations: ICI, immune checkpoint inhibitor; PFS, progression-free survival; CI, confidence interval; Sq, squamous cell carcinoma; PD, progressive disease; TPS, tumor proportion score.

**Table 1 cancers-17-01551-t001:** Patient characteristics.

		All Patients, *n* = 38
Age	Median (range), years	69.3 (37–83)
Histology (Sq/Non-Sq)	Non-Sq	24 (63.2)
Stage	Stage IV	32 (84.2)
	Recurrence after surgery	2 (5.3)
	Recurrence after CRT	4 (10.5)
Smoking	Current or past smoker	36 (94.7)
PS	0–1	38 (100)
Sex	Male	31 (81.6)
ICI retreatment regimen	Nivo	21 (55.2)
	Pembro	5 (13.2)
	Atezo	12 (31.6)
Prior chemo-ICI regimen	Platinum-doublet + Pembro	26 (68.4)
	Platinum-doublet + Nivo + Bev	1 (2.6)
	Platinum-doublet + Nivo + Ipi	1 (2.6)
	Platinum-doublet + Atezo ± Bev	6 (15.8)
	Carboplatin + PTX + radiation followed by durva	4 (10.5)
TPS *	<49%	26 (68.4)
	≥50%	9 (23.6)
CNS metastasis	Positive	12 (31.6)
Liver metastasis	Positive	4 (10.5)
Number of prior treatment lines	Median (range)	2.5 (1–4)
ICI-free interval	Median (range), months	11.9 (1.0–38.9)
Prior history of docetaxel		33 (86.8)
Prior chemo-ICI cessation	Due to PD	29 (76.3)
	Due to non-PD (toxicity/completion of durva)	8/1 (23.7)

Abbreviations: Sq, squamous; CRT, chemoradiotherapy; PS, performance status; ICI, immune checkpoint inhibitor; Nivo, nivolumab; Pembro, pembrolizumab; Atezo, atezolizumab; Bev, bevacizumab; Ipi, ipilimumab; Carboplatin, weekly carboplatin; PTX, weekly paclitaxel; Durva, durvalumab; TPS, tumor proportion score; CNS, central nervous system; PD, progressive disease. Data are presented as numbers (%) unless otherwise indicated. * Reported TPS was examined either before initial chemo-ICI or ICI retreatment. Three patients had no TPS data.

**Table 2 cancers-17-01551-t002:** ORR, DC*R*, and type of best response.

All Patients (*n* = 38)	
ORR, % (95% CI)	10.5 (2.9–24.8)
DCR, % (95% CI)	47.4 (31.0–64.2)
CR, n (%)	1 (2.6)
PR, n (%)	3 (7.9)
SD, n (%)	14 (36.8)
PD, n (%)	17 (44.7)
NA, n (%)	3 (7.9)
Patients with an ICI-free interval >11.9 months (*n* = 19)	
ORR, % (95%CI)	21.1 (6.1–45.6)
DCR, % (95%CI)	63.2 (38.4–83.7)
Patients with an ICI-free interval ≤ 11.9 months (*n* = 19)	
ORR, % (95%CI)	0 (0–17.6)
DCR, % (95%CI)	31.6 (12.6–56.6)

Abbreviations: ORR, overall response rate; CI, confidence interval; DCR, disease control rate; CR, complete response; PR, partial response; SD, stable disease; PD, progressive disease; NA, not assessed.

**Table 3 cancers-17-01551-t003:** Univariate and multivariate analysis of prognostic factors for PFS on ICI retreatment.

	Univariate Analysis		Multivariate Analysis	
	HR (95% CI)	*p*-Value	HR (95% CI)	*p*-Value
Histology: Non-Sq (vs. Sq)	0.40 (0.19–0.82)	0.013	0.64 (0.29–1.40)	0.26
ICI-free interval: >11.9 months (vs. ≤11.9 months)	0.42 (0.20–0.87)	0.019	0.33 (0.15–0.76)	0.009
Prior ICI: anti-PD-L1 (vs. PD-1)	0.41 (0.18–0.97)	0.042	0.33 (0.13–0.84)	0.020
Retreatment ICI: anti-PD-L1 (vs. PD-1)	1.85 (0.86–4.01)	0.12		
Liver metastasis: Positive	2.80 (0.96–8.15)	0.06		
Brain metastasis: Positive	1.02 (0.47–2.21)	0.96		
TPS ≥ 50% (vs. <50%)	0.85 (0.38–1.93)	0.70		
Age, years ≥ 75 (vs. <75)	0.89 (0.36–2.18)	0.80		
Prior ICI discontinuation due to non-PD (vs. PD)	0.83 (0.36–1.92)	0.66		
Serum Alb (g/L) ≥ 3.5	0.96 (0.48–1.92)	0.91		
Serum NLR ≥ 5	0.91 (0.45–1.84)	0.79		

Abbreviations: PFS, progression-free survival; ICI, immune checkpoint inhibitor; PD-1: programmed death 1; anti-PD-L1, anti-programmed death ligand 1; Sq, squamous cell carcinoma; TPS, tumor proportion score; Alb, albumin; NLR, neutrophil-to-lymphocyte ratio; HR, hazard ratio; CI, confidence interval.

**Table 4 cancers-17-01551-t004:** Adverse events.

	G1	G2	G3	G4	All Grade
Adverse events reported as irAEs (%)					
Colitis	0	2.6	0	0	2.6
Rash	2.6	2.6	2.6	0	7.9
Pneumonitis	0	2.6	0	0	2.6
Thyroid dysfunction	0	2.6	0	0	2.6
Myositis	0	0	2.6	0	2.6
Fever	5.3	0	0	0	5.3
Adverse events reported as non-irAEs (%)					
Anemia	2.6	2.6	0	0	5.3
Creatinine increased	0	2.6	0	0	2.6
Lung infection	0	2.6	0	0	2.6
AST increased	2.6	0	0	0	2.6
Hyperglycemia	0	2.6	0	0	2.6
Anorexia	0	2.6	0	0	2.6
Malaise	2.6	0	0	0	2.6
Shingles	0	2.6	0	0	2.6
AEs leading to ICI discontinuation: 2.6%					

Abbreviations: ICI, immune checkpoint inhibitor; G, grade; irAEs, immune-related adverse events.

## Data Availability

The data generated in this study are available upon request from the corresponding author.

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
