# Peer review of "Prospective Multi-Institutional Observational Study of Retreatment with Anti-PD-1/PD-L1 Antibodies in Patients with Non-Small Cell Lung Cancer Previously Treated with Anti-PD-1/PD-L1 Plus Chemotherapy: NJLCG (North Japan Lung Cancer Group) Trial 1901"

_cancers, 2025, doi:10.3390/cancers17091551_

Round 1
Reviewer 1 Report
Comments and Suggestions for Authors
- For patients with lung adenocarcinoma, is there data on the presence of mutations in genes such as EGFR, KRAS, ALK, etc.?
- What does the ICI-free interval depend on? It seems to me that this indicator is associated with the anti-PD-L1/anti-PD-1 therapy that was administered to patients, as well as with clinical and pathological characteristics. The longer the interval, the better the treatment outcome at the first stage, right? Maybe if we take another threshold value, we can identify a group with a better prognosis? In any case, this parameter cannot be influenced in any way.
- What determined which therapy the patient received, anti-PD-L1 or anti-PD-1?
- What determined the choice of a specific drug for therapy? Despite the fact that statistical differences between the drugs have not been shown, the response in the case of nivolumab was better.
- A longer ICI-free interval (than 11.9 months) and prior anti-PD-L1 treatment were identified as prognostic factors. It turns out that factors have been identified, none of which can be influenced. What is the significance of the information obtained for real clinical practice?
Reviewer 2 Report
Comments and Suggestions for Authors
This is a clinically meaningful and well-structured prospective study evaluating ICI monotherapy rechallenge in NSCLC patients previously treated with chemo-ICI. The topic is timely and relevant to current clinical practice. I appreciate the authors' efforts in gathering real-world data through a prospective design.
The ICI-free interval is highlighted as a prognostic factor for PFS. However, this interval includes patients who received various systemic treatments (e.g., chemotherapy) as well as those who were treatment-free. Since the presence or absence of treatment and the number of lines administered during this period could significantly influence PFS, I suggest adjusting for these factors as potential confounders in the multivariate analysis, or at least addressing their possible impact in the discussion.
The inclusion of a small number of patients who received durvalumab after chemoradiotherapy (CRT) may also introduce confounding, especially in the interpretation of ICI-free interval and prior PD-L1 therapy. It would strengthen the analysis if the authors could comment on whether the key findings remained consistent when these cases were excluded.
In the safety analysis, irAEs were captured only within 1 month of the final ICI dose. While this is acknowledged as a limitation, it would be helpful to further discuss how this short follow-up period might have contributed to an underestimation of irAE incidence, especially considering that delayed irAEs are not uncommon.
Regarding the objective response rate (ORR), the study was originally designed to detect an ORR of 20% with a threshold of 8%, requiring 63 patients. The actual ORR was 10.5% based on only 38 patients. This discrepancy suggests reduced statistical power and raises the possibility that a true treatment effect could have been missed (type II error). Further discussion of this issue, especially in relation to the wide confidence interval for ORR (2.9–24.8%), would improve the manuscript’s interpretability.
Finally, in Figure 4, a trend is noted regarding PFS benefit in patients receiving anti-PD-1 therapy after prior anti-PD-L1 treatment. However, no statistical test (e.g., log-rank) or p-value is shown. Please clarify whether statistical comparisons were conducted, and if so, consider including the p-value in the figure or figure legend.
Overall, this study contributes valuable insights into ICI rechallenge strategies. Addressing the above points would help enhance the clarity and impact of the findings.
Comments on the Quality of English LanguageThe manuscript is generally well written, there are a few instances where the English expression could be improved for clarity and fluency. A light revision by a native English speaker or professional editing service may further enhance the readability and ensure the scientific message is communicated clearly.
Round 2
Reviewer 1 Report
Comments and Suggestions for Authors
I have no more comments on the manuscript.
Author Response
Comments1:I have no more comments on the manuscript.
Reply: Thank you for taking the time to provide your valuable feedback on the revision. We are very grateful for your suggestions, which will help improve the quality of our research.
Reviewer 2 Report
Comments and Suggestions for Authors
Thank you very much for your thoughtful and comprehensive revisions addressing my previous comments. I appreciate the improvements made throughout the manuscript.
Previously, I suggested adding either an additional analysis or an expanded discussion regarding the potential influence of systemic treatments during the ICI-free interval.
While I appreciate the additional discussion you have provided, upon reflection, I believe that a statistical adjustment for treatment history (e.g., presence/absence of systemic chemotherapy, number of regimens) would have been preferable to strengthen the validity of the conclusions.
Although the current conclusion aligns with previous studies (such as WJOG9616L), further statistical exploration of the ICI-free interval would enhance the novelty and clinical applicability of your findings.
Thank you again for your careful revisions to the other points raised.
Author Response
Comments1: I believe that a statistical adjustment for treatment history (e.g., presence/absence of systemic chemotherapy, number of regimens) would have been preferable to strengthen the validity of the conclusions. Although the current conclusion aligns with previous studies (such as WJOG9616L), further statistical exploration of the ICI-free interval would enhance the novelty and clinical applicability of your findings.
Reply: Thank you for taking the time to provide your valuable feedback on the revision. Based on your feedback, I have added an exploratory analysis regarding the number of systemic treatment lines during the ICI-free period. There was no statistically significant trend observed between the number of prior treatment lines and PFS upon ICI retreatment in the group with a longer ICI-free interval. A point to mention is that in this study, patients in the group with a long ICI-free interval did not include those who were treatment-free during the interval; all patients had received at least one regimen of cytotoxic chemotherapy. This detail has been added to the manuscript (page 8, lines 240-244). The exploratory statistical analysis has been included as Figure S1.